

# Peripheral blood immune cell parameters in patients with high-grade squamous intraepithelial lesion (HSIL) and cervical cancer and their clinical value: a retrospective study

Ling Wang[1] and Yuyan Dong[2]

[1] Medical Department, University Hospital, Qingdao Agriculture University, Qingdao, China
[2] Department of Obstetrics and Gynecology, Key Laboratory of Birth Regulation and Control Technology of National Health Commission of China, Shandong Provincial Maternal and Child Health Care Hospital Affiliated to Qingdao University, Jinan, China

## ABSTRACT

**Objective**. The objective of this study was to delineate the profile of peripheral blood lymphocytic indices in patients afflicted with high-grade squamous intraepithelial lesions (HSIL) and cervical neoplasms, and to elucidate the correlation of these hematologic markers with the clinicopathological spectra in individuals diagnosed with cervical carcinoma.

**Methods**. We adopted a retrospective case-control modality for this investigation. An aggregate of 39 HSIL patients and 42 cervical carcinoma patients, who were treated in our facility from July 2020 to September 2023, were meticulously selected. Each case of cervical malignancy was confirmed through rigorous histopathological scrutiny. Concomitantly, 31 healthy female individuals, who underwent prophylactic health evaluations during the corresponding timeframe, were enlisted as the baseline control group. We systematically gathered and analyzed clinical demographics, as well as the neutrophil-to-lymphocyte ratio (NLR) and platelet-to-lymphocyte ratio (PLR), from peripheral blood samples. Pearson's correlation coefficient was deployed to dissect the interrelation between peripheral NLR and PLR concentrations and the clinicopathological features in the cervical cancer group.

**Results**. Inter-group comparative analysis unveiled statistically substantial variances in the PLR and NLR values among the tripartite clusters ($F = 36.941, 14.998, P < 0.001$, respectively). Although discrepancy in NLR ($P = 0.061$) and PLR ($P = 0.759$) measures between the groups of cervical carcinoma and HSIL was not statistically appreciable, these indices were markedly elevated in the cervical carcinoma faction as juxtaposed with the normative control group ($t = 5.094, 5.927; P < 0.001$ for both parameters). A discernible gradation in peripheral blood PLR and NLR concentrations was noted when stratified by clinical stage and the profundity of myometrial invasion in cervical cancer subjects ($P < 0.001$). The correlation matrix demonstrated a positive liaison between peripheral blood PLR and the clinical gradation, as well as the invasiveness of the neoplastic cells into the muscularis propria ($P < 0.05$); a similar trend was observed with the NLR values ($P < 0.05$).

Corresponding author
Yuyan Dong, dongyy2022@126.com

**Conclusion**. Augmented NLR and PLR levels in peripheral blood specimens are indicative of HSIL and cervical malignancy. These hematological parameters exhibit a pronounced interconnection with clinical staging and muscular wall penetration depth, serving as potential discriminative biomarkers for the diagnosis and prognosis of cervical cancer.

## INTRODUCTION

Cervical cancer is a common reproductive system disease in women, ranking fourth in incidence among female malignancies, with over 500,000 new cases globally each year (*Mohanty & Ghosh, 2015*). High-grade squamous intraepithelial lesion (HSIL) occurs in the transformation zone of the cervix. There is a close connection between tumor occurrence, development, and immune function (*Ungureanu et al., 2021*). Human papillomavirus (HPV) is the pathogen for cervical cancer and cervical squamous intraepithelial lesions. HPV infection triggers a series of immune responses, leading to abnormal cellular proliferation, poor differentiation, and aneuploidy, eventually progressing to precancerous lesions or malignant tumors (*Zang & Hu, 2021*). The tumor microenvironment (TME) refers to the local homeostatic environment closely related to tumor occurrence and development, which includes both the cancer cells and their surrounding milieu, encompassing the body's inflammatory responses and immune status (*Jayshree, 2021*). Increasing studies have shown that the characteristics of the tumor microenvironment are crucial factors affecting tumor growth, progression, migration, and treatment responses (*Lin et al., 2023*; *Regauer, Reich & Kashofer, 2022*). The neutrophil to lymphocyte ratio (NLR), as a reliable indicator to assess the status of individual immune inflammation, and the platelet to lymphocyte ratio (PLR), as an important biomarker reflecting the body's inflammatory response and immune function status, are closely associated with the occurrence and development of malignant tumours (*Diem et al., 2017*; *Li et al., 2018*). This study primarily investigates the levels of NLR and PLR in the peripheral blood of patients with HSIL and cervical cancer, assessing their relationship with the clinicopathological features of cervical cancer, thus providing a basis for the clinical diagnosis and treatment planning of cervical cancer.

## SUBJECTS AND METHODS

### Study population

In this rigorous retrospective case-control analysis, we meticulously enrolled 39 subjects presenting with HSIL and a group of 42 patients diagnosed with cervical carcinoma, all of whom received treatment within our institution from July 2020 through September 2023. Stringent histopathological confirmation was obtained for each case of cervical neoplasia. Exclusion criteria were robustly applied to omit individuals with concurrent neoplastic

entities, substantial organ pathologies, disturbances in endocrine metabolic homeostasis, or those who had partaken in immunosuppressive or antiviral pharmacotherapy in the proximity of the study commencement. Ensuring the integrity of the investigation, only patients with comprehensive clinical records and follow-up data were considered eligible. Parallelly, the study incorporated 31 healthy female participants, selected from those attending our hospital for routine health surveillance within the same temporal span. These participants were scrupulously screened to ensure the absence of any historical or current cervical neoplastic or precancerous conditions, chronic cervicitis, or ancillary gynecological reproductive afflictions.

The inclusion of subjects with HSIL in our study was motivated by the continuum of cervical neoplastic progression and the clinical need to explore the discriminatory potential of peripheral blood lymphocytic indices across the spectrum of cervical lesions. High-grade squamous intraepithelial lesions (HSIL) represent a precancerous stage in the transformation pathway of cervical neoplasms and are clinically significant due to their potential to progress to invasive cervical cancer. Given the close association between HSIL and the development of cervical cancer, investigating the peripheral blood neutrophil-to-lymphocyte ratio (NLR) and platelet-to-lymphocyte ratio (PLR) in patients afflicted with HSIL provides valuable insights into the hematologic alterations that may precede the onset of invasive malignancy. Moreover, the inclusion of HSIL patients allows for the examination of early hematologic changes that may serve as predictive biomarkers for disease progression, thereby offering a comprehensive assessment of the continuum from precancerous conditions to malignant tumors. By encompassing both HSIL and cervical carcinoma patients, our study seeks to elucidate the potential clinical value of peripheral blood NLR and PLR in discriminating the hematologic profiles associated with varying degrees of cervical neoplastic transformation.

All samples obtained in this study were approved by the ethics committee of the Shandong Provincial Maternal and Child Health Care Hospital and abided by the ethical guidelines of the Declaration of Helsinki, and ethics committee agreed to waive informed consent.

## Methods

Clinical data were collected from the patients, including (1) general information: age, menopausal status, number of pregnancies, number of deliveries, hypertension, diabetes, etc.; (2) laboratory indices: routine blood examination data including platelet count, absolute neutrophil count, and lymphocyte count. The NLR and PLR were calculated, with NLR being the ratio of neutrophil count to lymphocyte count, and PLR being the ratio of platelet count to lymphocyte count; (3) pathological data: cervical tissue specimens were processed for slide preparation and staining by pathologists at our institution, with diagnoses made under the microscope. Immunohistochemical staining results were incorporated when necessary. The pathological staging of cervical cancer followed the clinical staging criteria established by the International Federation of Gynecology and Obstetrics (FIGO) in 2018 (*Bhatla et al., 2018*).

## Statistical methods

The collated data from this investigative inquiry were subjected to rigorous statistical scrutiny utilizing the SPSS software, version 23.0. The graphical representations of the analyzed parameters were expertly generated *via* GraphPad Prism, version 8.0. The quantitative datasets were elucidated as means ± standard deviations. Continuous variables were first tested for normal distribution using the Shapiro–Wilk method. Dichotomous group comparisons were executed employing the Student's $t$-test, while the analysis of variance (ANOVA) with Bonferroni correction was the statistical method of choice for assessing differences across multiple groups. Categorical variables were succinctly described as frequencies and percentages, and their interrelations were interrogated using the Chi-square ($\chi^2$) test. Bonferroni correction was used for eliminating potential inflation of type I error due to multiple subgroup analyses. The Pearson correlation coefficient was the selected metric for the evaluation of correlational dynamics. A $P$-value threshold of less than 0.05 was pre-established to denote the threshold of statistical significance. Area under the receiver operating characteristic (AUC-ROC) was used to assess diagnostic value of PLR and NLR (*Mandrekar, 2010*). Furthermore, logistic regression analysis was conducted to evaluate the impact of various clinical and hematologic parameters on the occurrence of HSIL and cervical cancer. The coefficients for age, menopause, number of pregnancies, number of parturitions, hypertension, diabetes, PLR, and NLR were calculated, providing key insights into their contributions to the likelihood of HSIL or cervical cancer. Subsequently, a nomogram was constructed to visually present the results of the logistic regression analysis, facilitating a comprehensive depiction of the relative impact of each parameter on the occurrence of HSIL and cervical cancer. This enables a more intuitive interpretation of the relationships between the analyzed variables and the likelihood of disease occurrence.

## RESULTS

### Comparison of clinical data between cervical cancer, HSIL patients, and the healthy control group

Upon scrutinizing the demographic and clinical parameters across the three groups—individuals diagnosed with cervical carcinoma, patients identified with High-Grade Squamous Intraepithelial Lesions (HSIL), and the healthy control group—our statistical evaluation revealed a homogeneity in age distribution, menopausal status, and gravidity ($P$ >0.05), affirming comparability between groups as delineated in Table 1.

### Comparison of peripheral blood PLR and NLR levels among cervical cancer, HSIL patients, and the healthy control group

In the comparative analysis of systemic inflammatory indices, specifically the PLR and NLR in peripheral blood, as presented in Table 2, a statistically significant disparity was observed among the three groups ($P < 0.05$) (Figs. 1–2). While the intergroup comparison between cervical cancer and HSIL patients yielded no significant variance in NLR ($P = 0.061$) and PLR ($P = 0.759$) levels, a remarkable elevation in these biomarkers was noted in the cervical cancer group compared to the healthy controls, manifesting statistically

**Table 1 Comparison of clinical data among cervical cancer patients, HSIL patients, and healthy control group.**

| Parameter | Cervical cancer group ($n = 42$) | HSIL group ($n = 39$) | Healthy control group ($n = 31$) | $F/\chi^2$ | P |
|---|---|---|---|---|---|
| Age (years, $\overline{x} \pm s$) | $52.33 \pm 6.54$ | $51.89 \pm 7.08$ | $52.05 \pm 6.82$ | 0.043 | 0.958 |
| Menopause (cases/%) | | | | | |
| Yes | 18/42.86 | 16/41.03 | 12/38.71 | 0.127 | 0.939 |
| No | 24/57.14 | 23/58.97 | 19/61.29 | | |
| Number of pregnancies (mean $\pm$ SD) | $3.11 \pm 0.56$ | $2.97 \pm 0.61$ | $2.85 \pm 0.43$ | 2.056 | 0.133 |
| Number of deliveries (mean $\pm$ SD) | $2.35 \pm 0.27$ | $2.29 \pm 0.32$ | $2.17 \pm 0.35$ | 3.019 | 0.053 |
| Hypertension (cases/%) | | | | | |
| Yes | 9/21.43 | 8/20.51 | 5/16.13 | 0.346 | 0.841 |
| No | 33/78.57 | 31/79.49 | 26/83.87 | | |
| Diabetes (cases/%) | | | | | |
| Yes | 4/9.52 | 3/7.69 | 2/6.45 | 0.237 | 0.888 |
| No | 38/90.48 | 36/92.31 | 29/93.55 | | |

Notes.

HSIL, high-grade squamous intraepithelial lesion; F, the F-statistic used for analysis of variance (ANOVA) to assess differences across multiple cohorts; $\chi^2$, chi-square test.

**Table 2 Comparison of peripheral blood NLR and PLR among cervical cancer patients, HSIL patients, and healthy control group (mean $\pm$ SD).**

| Parameter | Cervical cancer group ($n = 42$) | HSIL group ($n = 39$) | Healthy control group ($n = 31$) | F | P |
|---|---|---|---|---|---|
| PLR | $179.57 \pm 35.46$ | $174.15 \pm 36.89$ | $141.28 \pm 25.82$ | 36.941 | <0.001 |
| NLR | $2.89 \pm 0.44$ | $2.74 \pm 0.82$ | $2.03 \pm 0.79$ | 14.998 | <0.001 |

Notes.

HSIL, high-grade squamous intraepithelial lesion; F, the F-statistic used for analysis of variance (ANOVA) to assess differences across multiple cohorts.

significant deviations ($t = 5.094, 5.927$; $P < 0.001$ for PLR and $t = 4.206, 3.657$; $P < 0.001$ for NLR).

## Comparison of peripheral blood PLR levels in cervical cancer patients with different pathological features

In evaluating the PLR among cervical cancer patients with differing pathological features, our analysis revealed significant variations in PLR values associated with clinical stage and degree of muscle wall invasion (Table 3). Specifically, PLR values increased with advancing clinical stage, being $150.88 \pm 49.53$ for Stage I, $175.36 \pm 33.52$ for Stage II, and $193.15 \pm 25.81$ for Stage III, with a $t$ value of 3.422, indicating a statistically significant difference ($P = 0.043$). Similarly, a significant association was identified between PLR and the degree of muscle wall invasion, with PLR values of $151.29 \pm 39.48$ for 2/3 to full-thickness invasion, $171.53 \pm 30.57$ for 1/2 to 2/3 thickness invasion, and $189.00 \pm 26.83$ for <1/2 thickness invasion, achieving an F value of 3.510 ($P = 0.040$). In contrast, comparisons based on tumor diameter (<four cm: $173.15 \pm 31.63$ *vs.* $\geq 4$ cm: $179.89 \pm 32.37$), presence of vascular invasion (present: $176.15 \pm 38.19$ *vs.* absent: $174.69 \pm 36.22$), and nerve invasion

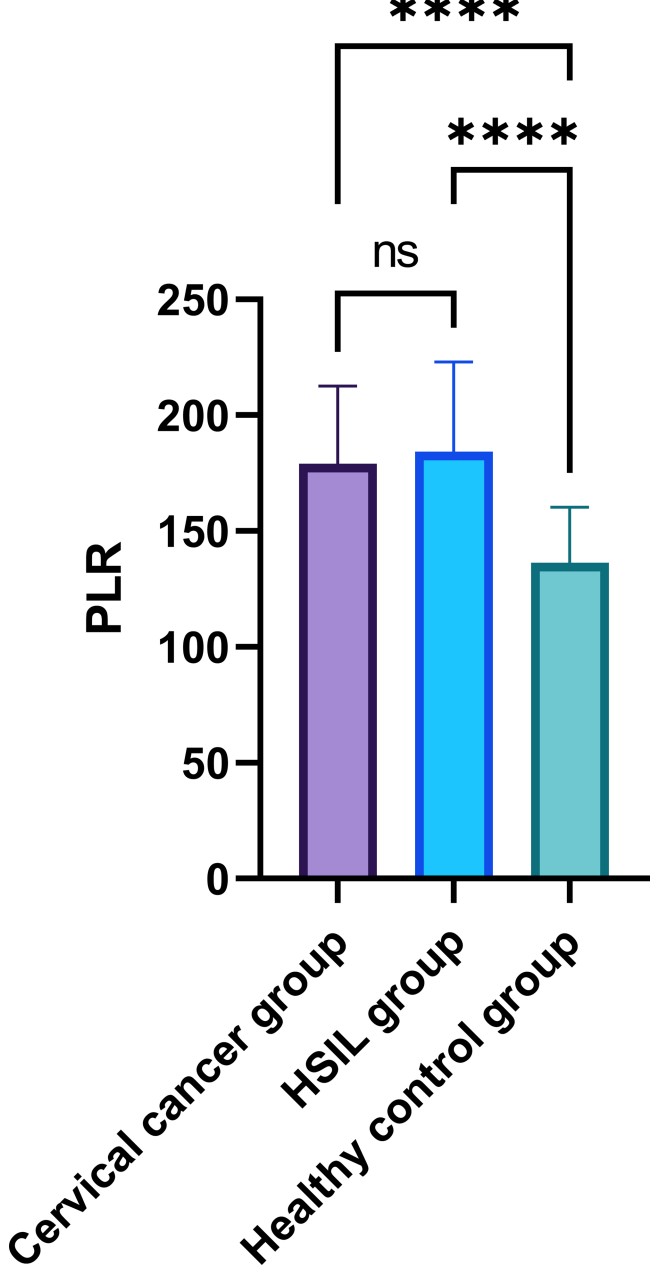

**Figure 1** **Comparison of peripheral blood PLR among patients with cervical cancer, HSIL, and healthy control group.** Note: ∗∗∗∗: $P < 0.0001$; ns: $P > 0.05$.

(present: $179.32 \pm 38.45$ *vs.* absent: $177.07 \pm 37.42$) did not yield statistically significant differences, with $P$-values of 0.510, 0.903, and 0.863, respectively. These results highlight the significance of PLR variations in relation to clinical stage and the extent of muscle wall invasion in cervical cancer patients.

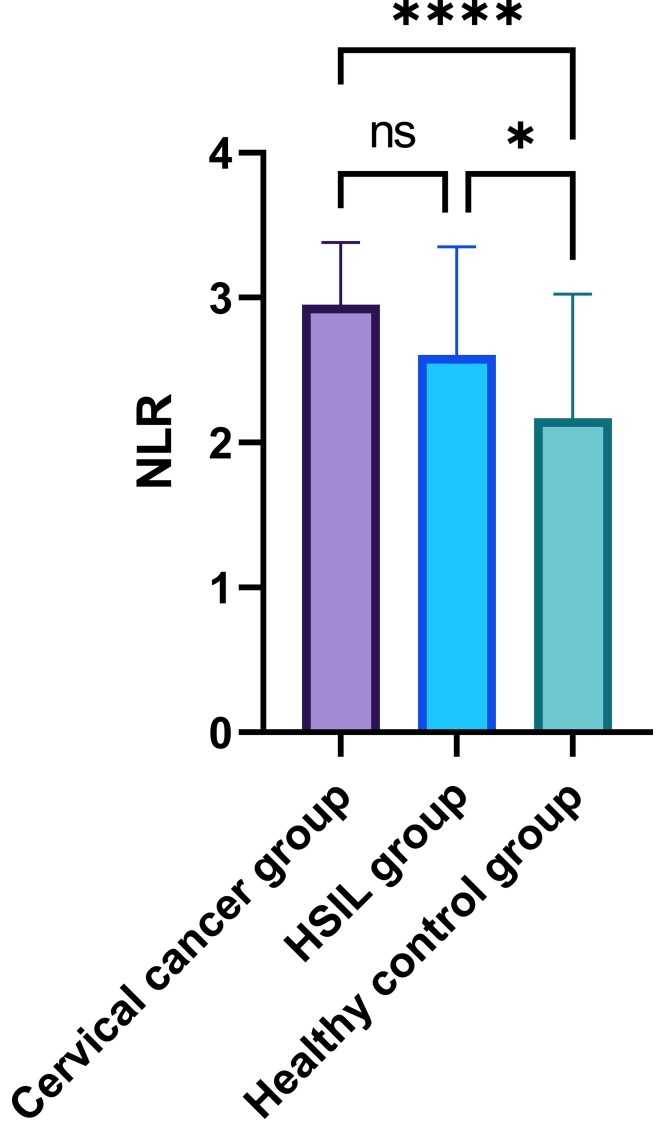

**Figure 2 Comparison of peripheral blood NLR among patients with cervical cancer, HSIL, and healthy control group.** Note: ∗∗∗∗: $P < 0.0001$; ns: $P > 0.05$.

## Comparison of peripheral blood NLR levels in cervical cancer patients with different pathological features

In the assessment of the NLR in patients with cervical cancer based on different pathological features, significant variations were observed across clinical stages (Table 4). Specifically, Stage I patients exhibited an NLR of $1.95 \pm 0.44$, which significantly increased in Stage II to $2.23 \pm 0.38$, and further to $2.94 \pm 0.31$ in Stage III, with an F value of 8.267 ($P = 0.001$),

**Table 3 Comparison of peripheral blood PLR in cervical cancer patients with different pathological features ($n = 42$).**

| Pathological feature | Number of cases | PLR ($\bar{x} \pm s$) | t/F | P |
|---|---|---|---|---|
| Clinical stage | | | | |
| Stage I | 12 | 150.88 ± 49.53 | 3.422 | 0.043 |
| Stage II | 21 | 175.36 ± 33.52 | | |
| Stage III | 9 | 193.15 ± 25.81 | | |
| Tumor diameter | | | | |
| <4 cm | 26 | 173.15 ± 31.63 | 0.665 | 0.510 |
| ≥4 cm | 16 | 179.89 ± 32.37 | | |
| Degree of muscle wall invasion | | | | |
| 2/3 to full-thickness | 17 | 151.29 ± 39.48 | 3.510 | 0.040 |
| 1/2 to 2/3 thickness | 22 | 171.53 ± 30.57 | | |
| <1/2 thickness | 3 | 189.00 ± 26.83 | | |
| Vascular invasion | | | | |
| Present | 15 | 176.15 ± 38.19 | 0.123 | 0.903 |
| Absent | 27 | 174.69 ± 36.22 | | |
| Nerve invasion | | | | |
| Present | 11 | 179.32 ± 38.45 | 0.174 | 0.863 |
| Absent | 31 | 177.07 ± 37.42 | | |

**Notes.**

PLR, platelet-to-lymphocyte ratio; F, the F-statistic used for analysis of variance (ANOVA) to assess differences across multiple cohorts; t, Student's t-test.

indicating a statistically significant difference in NLR values across these stages. Conversely, when comparing NLR based on tumor diameter (<four cm: 2.54 ± 0.57 *vs.* ≥4 cm: 2.77 ± 0.58), invasion depth (2/3 to full thickness: 1.96 ± 0.45 *vs.* 1/2 to 2/3 thickness: 2.30 ± 0.38 *vs.* <1/2 thickness: 2.93 ± 0.27), presence of vascular invasion (present: 2.94 ± 0.57 *vs.* absent: 2.83 ± 0.61), and presence of neural invasion (present: 2.92 ± 0.53 *vs.* absent: 2.79 ± 0.58), no statistically significant differences were found (*P*-values of 0.214 for tumor diameter, <0.001 for invasion depth, 0.570 for vascular invasion, and 0.518 for neural invasion). These findings underscore the significant association between NLR and the clinical stage of cervical cancer, while the correlations with tumor diameter, invasion depth, and presence of vascular or neural invasions did not exhibit statistical significance.

## Correlation analysis of peripheral blood PLR and NLR with pathological features in cervical cancer patients

In the correlation analysis between peripheral blood markers and pathological features in patients with cervical cancer, the PLR and NLR were evaluated (Table 5). The results revealed a statistically significant positive correlation between PLR and clinical stage ($r = 0.459$, $P = 0.013$), as well as a similar significant correlation between NLR and clinical stage ($r = 0.439$, $P = 0.020$). Additionally, a significant correlation was observed between both PLR ($r = 0.432$, $P = 0.029$) and NLR ($r = 0.417$, $P = 0.031$) with invasion depth. Conversely, correlations between PLR and NLR with pathological type, tumor diameter, and the presence of vascular or neural invasion did not reach statistical significance. Specifically,

**Table 4 Comparison of peripheral blood NLR in patients with cervical cancer based on different pathological features (n = 42).**

| Pathological feature | Number of cases | NLR ($\bar{x} \pm s$) | t/F | P |
|---|---|---|---|---|
| Clinical stage | | | | |
| Stage I | 12 | 1.95 ± 0.44 | 8.267 | 0.001 |
| Stage II | 21 | 2.23 ± 0.38 | | |
| Stage III | 9 | 2.94 ± 0.31 | | |
| Tumor diameter | | | | |
| <4 cm | 26 | 2.54 ± 0.57 | 1.262 | 0.214 |
| ≥4 cm | 16 | 2.77 ± 0.58 | | |
| Invasion depth | | | | |
| 2/3 to full thickness | 17 | 1.96 ± 0.45 | 8.494 | <0.001 |
| 1/2 to 2/3 | 22 | 2.30 ± 0.38 | | |
| <1/2 | 3 | 2.93 ± 0.27 | | |
| Vascular invasion | | | | |
| Present | 15 | 2.94 ± 0.57 | 0.573 | 0.570 |
| Absent | 27 | 2.83 ± 0.61 | | |
| Neural invasion | | | | |
| Present | 11 | 2.92 ± 0.53 | 0.652 | 0.518 |
| Absent | 31 | 2.79 ± 0.58 | | |

Notes.

NLR, neutrophil-to-lymphocyte ratio; F, the F-statistic used for analysis of variance (ANOVA) to assess differences across multiple cohorts; $t$, Student's $t$-test.

**Table 5 Correlation analysis between peripheral blood PLR, NLR, and pathological features in patients with cervical cancer.**

| Parameter | PLR | | NLR | |
|---|---|---|---|---|
| | r | P | r | P |
| Pathological type | 0.058 | 0.616 | 0.218 | 0.141 |
| Clinical stage | 0.459 | 0.013 | 0.439 | 0.020 |
| Tumor diameter | 0.244 | 0.197 | 0.263 | 0.165 |
| Invasion depth | 0.432 | 0.029 | 0.417 | 0.031 |
| Vascular or neural invasion | 0.031 | 0.823 | 0.044 | 0.869 |

Notes.

PLR, platelet-to-lymphocyte ratio; NLR, neutrophil-to-lymphocyte ratio; r, the Pearson correlation coefficient.

PLR's correlations with pathological type ($r = 0.058$, $P = .616$), tumor diameter ($r = 0.244$, $P = 0.197$), and vascular or neural invasion ($r = 0.031$, $P = 0.823$) exhibited $P$-values well above the threshold for significance. Similarly, NLR's correlations with these parameters— pathological type ($r = 0.218$, $P = 0.141$), tumor diameter ($r = 0.263$, $P = 0.165$), and vascular or neural invasion ($r = 0.044$, $P = 0.869$)—also did not demonstrate statistical significance. These findings suggest that both PLR and NLR are significantly associated with the clinical stage and invasion depth of cervical cancer, highlighting their potential as indicators of disease progression.

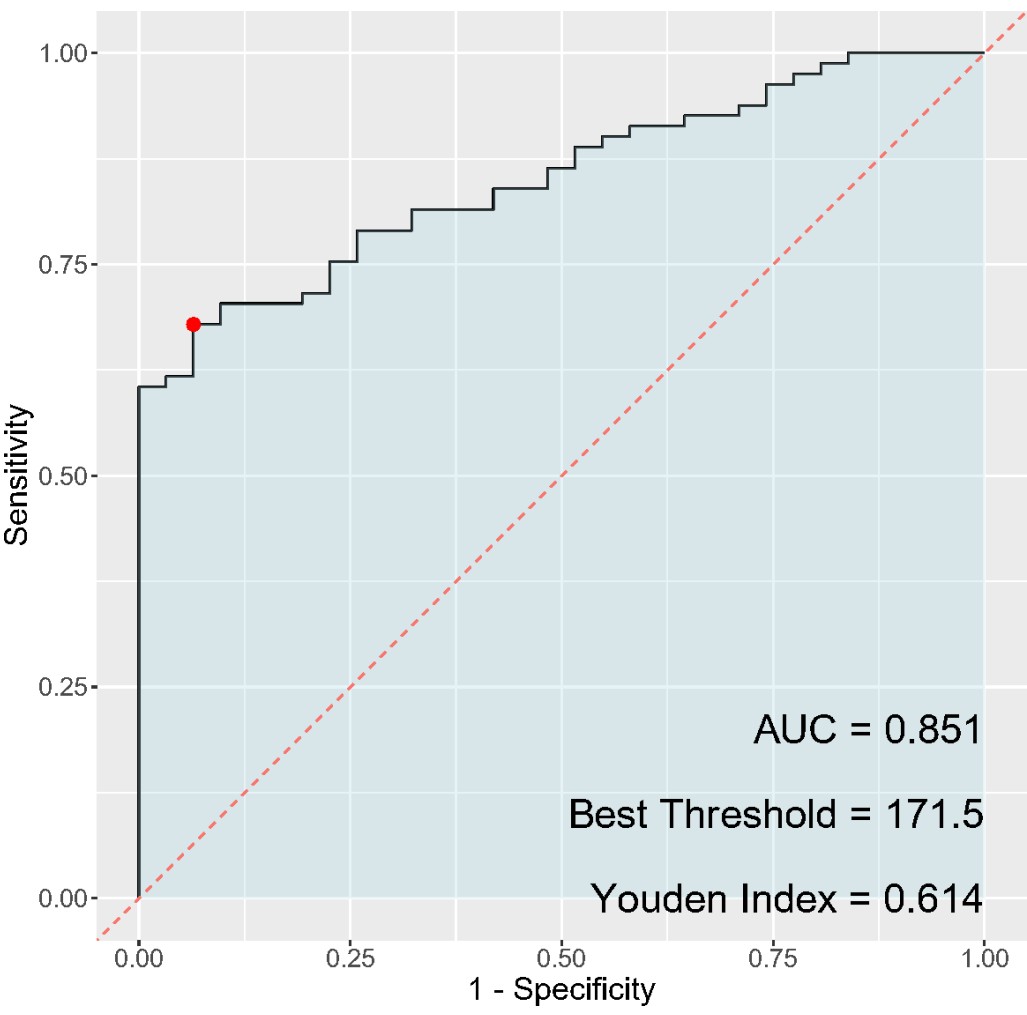

**Figure 3** ROC curve of PLR for cervical cancer diagnosis.

## ROC analysis of PLR and NLR and the occurrence of cervical cancer and HSIL

In the analysis of inflammatory markers' discriminatory ability for cervical cancer diagnosis, the PLR (Fig. 3) exhibited a sensitivity of 67.9% and a specificity of 93.5%, while the NLR (Fig. 4) had a sensitivity of 65.4% and a specificity of 77.4%. The AUC for PLR was determined to be 0.851, indicating a superior diagnostic performance compared to the NLR, which presented an AUC of 0.734. Regarding the Youden index, a measure used to assess the effectiveness of a diagnostic test, PLR obtained a score of 0.614, thereby outperforming NLR, which had a Youden index of 0.428. Although both PLR and NLR demonstrated statistical significance in their diagnostic capacities, the results emphasize a notably higher specificity and overall diagnostic accuracy for PLR, suggesting its potential greater utility as a biomarker for cervical cancer compared to NLR.

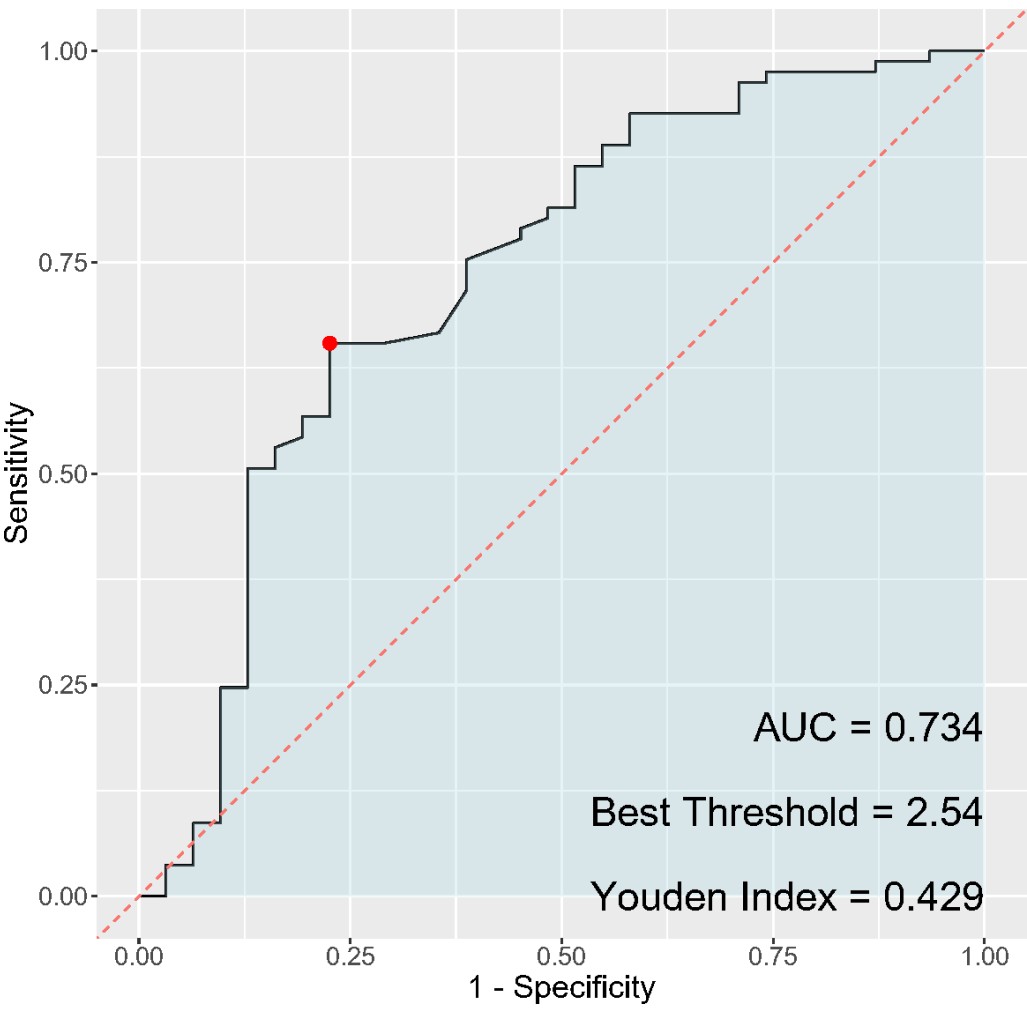

**Figure 4** ROC curve of NLR for cervical cancer diagnosis.

## Logistic regression analysis of various parameters and the occurrence of cervical cancer and HSIL

In the logistic regression analysis, the coefficients for age, menopause, number of pregnancies, number of parturitions, hypertension, and diabetes were 0.01, 0.136, 0.846, 0.479, 0.323, and 0.316, respectively (Table 6). The odds ratios for these variables were 1.011, 1.145, 2.331, 1.615, 1.381, and 1.372, suggesting the impact of each unit increase in the predictor variables on the likelihood of HSIL or cervical cancer. The 95% confidence intervals (CI) for the odds ratios provided additional insight into the precision of the estimates. Notably, the odds ratio for the number of pregnancies demonstrated a higher value, with an upper confidence interval greater than 1, indicating a potentially significant impact on the likelihood of HSIL or cervical cancer. Similarly, the odds ratios for hypertension, diabetes, and PLR were also noteworthy, emphasizing a potential association with the risk of HSIL or cervical cancer. Moreover, the analysis revealed a statistically significant association of neutrophil-to-lymphocyte ratio (NLR) with the presence of HSIL

**Table 6** Logistic regression analysis of various parameters and the occurrence of cervical cancer and HSIL.

| Parameter | Age | Menopause | Number of pregnancies | Number of parturition | Hypertension | Diabetes | PLR | NLR |
|---|---|---|---|---|---|---|---|---|
| Coef | 0.01 | 0.136 | 0.846 | 0.479 | 0.323 | 0.316 | 0.044 | 1.212 |
| Odds ratio | 1.011 | 1.145 | 2.331 | 1.615 | 1.381 | 1.372 | 1.045 | 3.362 |
| Lower CI | 0.949 | 0.495 | 1.182 | 0.649 | 0.487 | 0.31 | 1.028 | 1.809 |
| Upper CI | 1.076 | 2.722 | 4.888 | 4.394 | 4.549 | 9.57 | 1.067 | 6.775 |
| B | 0.33 | 0.314 | 2.354 | 0.997 | 0.577 | 0.38 | 4.811 | 3.634 |
| P Value | 0.741 | 0.753 | 0.019 | 0.319 | 0.564 | 0.704 | $p < 0.001$ | $p < 0.001$ |

**Notes.**

PLR, platelet-to-lymphocyte ratio; NLR, neutrophil-to-lymphocyte ratio; r, the Pearson correlation coefficient.

or cervical cancer, as evidenced by a $P$ value of less than 0.001. The regression coefficient (B) values for PLR and NLR were 0.38 and 4.811, respectively, underlining the potential role of these hematologic parameters as discriminative biomarkers for the diagnosis of HSIL and cervical cancer. Finally, the results of logistic regression analysis were visualized by nomogram (Fig. 5).

## DISCUSSION

Neoplasms remains the main killer worldwide (*Hu et al., 2016*; *Sun et al., 2021*; *Li & Qiao, 2022*; *Ma et al., 2016*; *Yang et al., 2019*), among which, cervical cancer is one of the most common malignant diseases in gynecology, and its development from normal cervical changes to precancerous lesions such as HSIL and eventually to cervical cancer is often underpinned by persistent infection with high-risk types of HPV. Previous research has indicated that viral infections can lead to changes in inflammatory cells within the bloodstream, and chronic inflammation is recognized as one of the biological hallmarks of malignant tumors (*Alimena et al., 2022*). Besides enhancing immune responses, inflammation can also induce immunosuppression, leading to genetic mutations, increased cell cycle progression, and aberrant cellular proliferation during repair processes. These pathophysiological responses can facilitate the onset and progression of tumors. Furthermore, cytokines released during chronic inflammation can induce angiogenesis, alter the expression of oncogenes and tumor suppressor genes, and inhibit apoptosis, resulting in abnormal inflammatory signalling pathways; chronic inflammation also promotes the establishment of an immunosuppressive tumor microenvironment (TME) (*Santella et al., 2022*; *Yang et al., 2021*). Although both immunity and inflammation are fundamental characteristics of the TME, the activation of systemic immune responses by malignant tumors allows the detection of inflammatory cells surrounding tumor cells, suggesting that blood inflammatory cells can serve as biomarkers for the diagnosis of malignant tumors (*Franciosi et al., 2022*).

Neutrophils, platelets, and lymphocytes are peripheral blood cell indices that can act as markers for the progression of inflammatory diseases. The PLR is a commonly used indicator of inflammation and a measure of disease severity. In states of inflammation, platelet counts can increase while lymphocyte counts may decrease, typically resulting in an

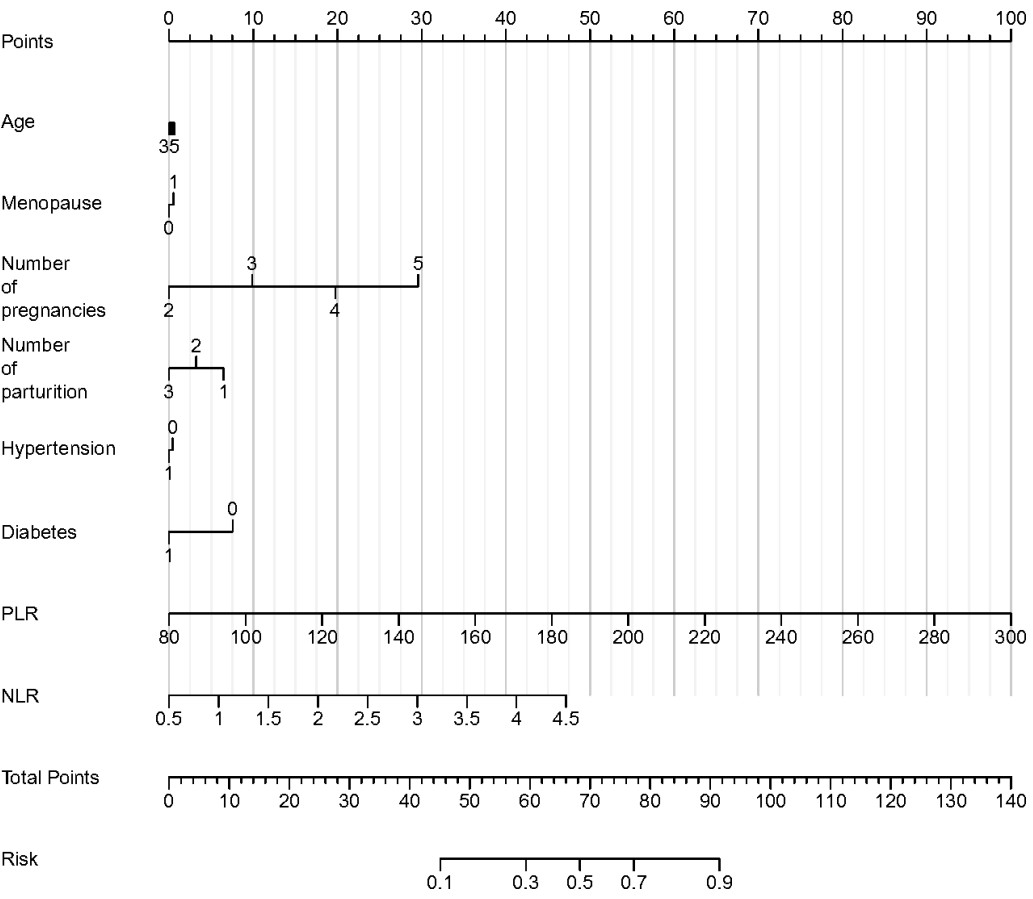

**Figure 5** Nomogram of various parameters and the occurrence of cervical cancer and HSIL.

elevated PLR as inflammation leads to platelet aggregation and lymphocyte reduction (*Trinh et al., 2020*). Literature has shown that a high PLR level is closely associated with the clinicopathological characteristics and poor prognosis of patients with cervical cancer (*Zhu et al., 2018*). In our study, the PLR levels in the peripheral blood of the cervical cancer group were significantly higher than those in the healthy control group, indicating that PLR levels in patients with HSIL and cervical cancer are further elevated when compared to the healthy population. Moreover, the comparison of peripheral blood PLR levels in cervical cancer patients with different clinical stages and degrees of muscular wall infiltration showed statistically significant differences. Further correlation analysis revealed a positive relationship between peripheral blood PLR in cervical cancer patients and clinical staging as well as the degree of muscular wall infiltration, suggesting that higher PLR levels are closely associated with the clinicopathological features of cervical cancer patients, with more severe conditions correlating with higher PLR levels.

The NLR is the ratio of the number of neutrophils to lymphocytes in the blood, often used as a metric for assessing disease severity, prognostication, and guiding therapy. Under inflammatory or infectious conditions, the counts of neutrophils and lymphocytes typically change; neutrophils play a central role in inflammatory and infection responses,

participating in the inflammatory process and immune cell killing functions. Lymphocytes are an essential component of the immune response, generally increasing following infection (*Tas et al., 2019*). Past research has confirmed that an increase in NLR may indicate higher levels of inflammation, decreased immune function, or stress response in the body, and higher NLR values are often associated with poor prognosis in malignant tumors (*Salzano et al., 2021*; *Cheng et al., 2022*). The results of our study indicate that the NLR levels in the peripheral blood of the cervical cancer group were significantly higher than those in the healthy control group, suggesting that compared to the healthy population, patients with HSIL and cervical cancer have further elevated peripheral blood NLR levels. Correlation analysis revealed that peripheral blood NLR in cervical cancer patients was positively related to clinical stage and degree of muscular wall infiltration, indicating a close association between higher NLR levels and the clinicopathological features of cervical cancer patients, consistent with previous literature reports (*Zou, Yang & Li, 2020*). The speculated reason might be that chronic inflammation occurs repeatedly in the cervix under persistent HPV infection, leading to an increase in neutrophils and a decrease in lymphocytes under the regulation of the body's inflammatory system, causing disarray in the body's system and facilitating tumor cell escape. Additionally, further infiltration of inflammatory cells releases harmful substances like oxygen free radicals and nitric oxide (NO) that can damage normal cells, potentially causing normal cells to mutate into malignant tumor cells. These inflammatory factors can also damage the endothelial cells of blood vessels, activate the blood coagulation system, and increase platelets, thereby leading to elevated levels of peripheral blood PLR and NLR.

The findings of our study align with and contribute to the body of knowledge on the role of peripheral blood PLR and NLR as potential indicators of disease severity and progression in cervical cancer (*Huang et al., 2019*; *Prabawa et al., 2019*). Previous studies (*Huang et al., 2019*; *Prabawa et al., 2019*; *Li et al., 2021*) have also reported an association between elevated PLR and NLR levels and adverse clinicopathological features in various malignancies, including cervical cancer. Our results corroborate and extend these findings, providing further evidence of the clinical relevance of these hematologic parameters in the context of cervical malignancies. Moreover, the observed positive correlations between PLR, NLR, clinical stage, and invasion depth are consistent with similar investigations (*Diem et al., 2017*; *Nøst et al., 2021*) in other cancer types, highlighting the systemic nature of inflammation and immune response in tumor microenvironments.

Comparisons with previous studies (*Prabawa et al., 2019*; *Gawiński, Hołdakowska & Wyrwicz, 2022*; *Ozmen et al., 2017*) underscore the utility of PLR and NLR as non-invasive biomarkers for assessing disease severity, guiding treatment decisions, and predicting outcomes in cervical cancer. The establishment of these correlations further strengthens the potential clinical value of PLR and NLR in aiding risk stratification, treatment selection, and patient prognostication. Furthermore, the observed correlations between PLR, NLR, and specific clinicopathological features provide insights into the interplay between systemic inflammation and tumor progression in cervical cancer. While further research is warranted to elucidate the mechanistic underpinnings of these associations, our study

contributes to the growing body of evidence supporting the use of PLR and NLR as readily accessible and informative biomarkers in the clinical management of cervical cancer.

Our study represents a preliminary exploration of the potential clinical value of peripheral blood PLR and NLR as biomarkers in cervical cancer. While the results obtained from the correlation analysis provide early indicators of the associations between PLR, NLR, and specific clinicopathological features, it is crucial to acknowledge that further research is essential to validate and extrapolate these findings. Additionally, the complexities of the tumor microenvironment and the multifactorial nature of cervical cancer necessitate comprehensive profiling and longitudinal assessments to establish the true predictive and prognostic utility of PLR and NLR in clinical practice. Moreover, larger group studies and multivariate analyses are required to elucidate the independent contributions of PLR and NLR to disease progression, patient outcomes, and treatment responses.

In consideration of the limitations of our study, it is imperative to acknowledge the relatively small sample size and the potential impact on the generalizability of the results. While the findings provide important insights into the potential clinical value of peripheral blood NLR and PLR in the context of cervical neoplasms, the modest sample size necessitates caution in extrapolating the results to broader populations. Besides, an important consideration in interpreting the findings of this study is the limitation related to the depth of immunophenotypic characterization and the assessment of co-populations involving NK cells. While the current investigation provided valuable insights into the hematological parameters and their association with HSIL and cervical cancer, it is acknowledged that a detailed analysis of the immunophenotype and the specific co-populations of NK cells, which have been extensively examined in the literature, was not carried out. It has been demonstrated in previous research, as indicated by *Konjevic et al. (2012)*, that a comprehensive examination of the immunophenotype and co-populations of NK cells can provide more nuanced and specific indicators of immune system alterations in the context of tumor development and progression.

Although our results serve as early indicators and provide initial insights into the potential implications of PLR and NLR in cervical cancer, a cautious approach is warranted in drawing definitive conclusions. The dynamic and evolving nature of cancer biomarker research necessitates continuous validation and refinement of findings to ensure their clinical applicability. As such, our study lays the foundation for future investigations to build upon, fostering a deeper understanding of the intricate interplay between systemic inflammation, immune status, and the pathophysiology of cervical cancer. The accumulation of robust evidence over time will ultimately determine the translational significance of peripheral blood PLR and NLR as adjunctive tools in the comprehensive management of cervical cancer.

## CONCLUSION

Patients with HSIL and cervical cancer have elevated levels of peripheral blood NLR and PLR, and the levels of NLR and PLR in peripheral blood of cervical cancer patients are closely associated with clinical staging and the degree of muscular wall infiltration,

which can aid in the differential diagnosis of cervical cancer. This study is a single-center retrospective analysis with a relatively small sample size, which may introduce some bias to the results. Future research should include a larger number of cases and combine multiple indicators to analyze the diagnostic efficacy for cervical cancer and to explore the mechanisms of action.

### Funding
The authors received no funding for this work.

### Competing Interests
The authors declare there are no competing interests.

### Author Contributions
- Ling Wang conceived and designed the experiments, performed the experiments, analyzed the data, prepared figures and/or tables, authored or reviewed drafts of the article, and approved the final draft.
- Yuyan Dong conceived and designed the experiments, performed the experiments, analyzed the data, prepared figures and/or tables, authored or reviewed drafts of the article, and approved the final draft.

### Human Ethics
The following information was supplied relating to ethical approvals (*i.e.*, approving body and any reference numbers):

All samples obtained in this study were approved by the ethics committee of the Shandong Provincial Maternal and Child Health Care Hospital and abided by the ethical guidelines of the Declaration of Helsinki.

### Data Availability
The raw data is available in the Supplementary File.

### Supplemental Information
Supplemental information for this article can be found online at http://dx.doi.org/10.7717/peerj.17499#supplemental-information.

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
