# Peer review of "Peripheral blood immune cell parameters in patients with high-grade squamous intraepithelial lesion (HSIL) and cervical cancer and their clinical value: a retrospective study"

_PeerJ, doi:10.7717/peerj.17499_

## Round 0.1 · original submission · Major Revisions

Based on the answers of the two reviewers, the decision is that the work requires correction.

Reviewer 1 ·

Basic reporting

In the submitted manuscript authors assessed peripheral blood lymphocyte parameters in patients with high-grade squamous intraepithelial lesion (HSIL) and cervical cancer (CC) and their clinical value.
Unfortunately, the quality of this manuscript is not too high, since the sample size is relatively small and statistical analyses were not properly applied, obtained results were not properly interpreted, while there are lots of previous such studies which have not been mentioned and compared to.

Experimental design

Since authors had relatively small sample size, ad hoc using of parametric statistical tests is not admissible. Therefore, authors should first test normality of distribution of numerical variables and accordingly choose statistical tests. Furthermore, a post hoc test used for multi-group comparisons should be stated, correlation coefficients must be properly interpreted (see section 3. Validity of the findings), since there are many sub-group analyses some kind of p-value adjustment, like for example Bonferroni's, must be applied, while diagnostic value of those two measured peripheral blood lymphocyte parameters could easily be assessed by calculating area under the receiver operating characteristic (AUC-ROC) (see PMID: 20736804 for proper interpretation of AUC).

Validity of the findings

There are main problems with this manuscript because of which the validity of findings presented in it are misleading or unreliable:
1. Since the precancerous lesion for invasive adenocarcinoma of cervix is adenocarcinoma in situ and not HSIL, authors could not include those eight cases of adenocarcinoma in CC group.
2. Authors should carefully inspect PMID: 23638278 and adequately interpret their statistically significant correlation coefficients because they put too high significance on pretty low correlations.
3. Since the sample size is relatively low and there are many sub-group analyses, some kind of p-value adjustment must be applied and the results of statistical analyses (p-values) properly interpreted.

Additional comments

Table 2 should precede Figure 1 and 2 since they all in essence presented the same things, while on those figures statistically significantly different groups should be somehow marked.
All abbreviations presented in figures and tables must be explained in legends and footnotes, respectively.
When a single result of statistical analysis is being presented in 'Abstract' or main text, the exact p-value should be provided.
Correlation coefficient must always be stated with its p-value (and properly interpreted).
'Discussion' is too scarce and must include comparisons with previous such studies, which are really abundant.
Explanations of abbreviations like HSIL, PLR and NLR should not be written with capitalized first letters.

Reviewer 2 ·

Basic reporting

Basic reporting of this paper is satisfactory. Language is clear and references sufficient.

Experimental design

In my opinion, the statistical analysis needs to be refined and reworked upon and would suggest a statistical consultation.

Validity of the findings

Described in additional comments.

Additional comments

• In my opinion, the study is still very early to be considered for publication, results that authors have tabulated are just early indicators, are very crude and do not lead to any conclusion.

• Authors need strong statistical support to execute and present this study.

• Authors have claimed this to be a ‘retrospective case-control’ study. However, the term ‘cohort’ has been used in a few instances. Please confirm if the authors meant this to be a case-control or a cohort study and use the terms appropriately.

• If authors wanted to study the PLR and NLR status in relation to cervical cancer (as stated in the objective of the paper), the rationale to include subjects with HSIL is not clear.

• Terms such as ‘F/X2’, ‘F’, ‘t/F’ , ‘r’ etc lacks clarity and needs to be described in the “Notes” or ‘Legend” of corresponding tables.

• In my opinion a multivariate analysis is required to assess the relationship between study parameters (NLR and PLR) and disease itself, stage of disease and additional variable like muscular wall invasion- consult a statistical expert.

• Correlation coefficients calculated from the Pearson correlation are not provided (if authors want to limit their analysis to correlations.)

---

## Round 0.2 · Minor Revisions

Changes are needed

the title of the paper suggests that it is an examination of lymphocyte populations. Modern methods use a flow cytometer to determine individual populations of B, T, and NK cells, which was not done here.

Therefore, the title should not mislead the readers into thinking that the immunophenotyping of the cells was performed.
The title should contain exactly what was tested, which is the LN ratio, which is very limited compared to testing using a large panel of monoclonal antibodies.
In addition, a big problem is that only references from authors from China and self-citations are cited, as if no one else had examined lymphocyte populations in tumors.

Papers on NLR and other relationships in SLE, Neurological and other diseases by other authors from Europe were not taken into consideration or discussed, and should be added.

In the limitation of the study, it is necessary to indicate that a complex examination of the immunophenotype and especially of the co-population of NK cells, which would give better indicators of changes in the immune system in tumors, was not carried out, which was examined in many papers and which was also shown earlier in papers PMID: 22442005

---

## Round 0.3 · Minor Revisions

Please correct the work according to the final requests of the reviewer

Reviewer 1 ·

Basic reporting

no comment

Experimental design

Logistic regression analysis and nomogram creation must be explained in "1.3 Statistical Methods" section.

Validity of the findings

no comment

Additional comments

Although I could not obtain/find authors' responses to my comments, I observed that they have implemented them in the revised manuscript.

---

## Round 0.4 · accepted · Accept

I am writing to inform you that your manuscript - Peripheral blood immune cell parameters in patients with high-grade squamous intraepithelial lesion (HSIL) and cervical cancer and their clinical value: a retrospective study - has been Accepted for publication.

Reviewer 1 ·

Basic reporting

no comment

Experimental design

no comment

Validity of the findings

no comment

Additional comments

no comment